

**Altitude Registration of Limb-Scattered Radiation**
Leslie Moy[1], P.K. Bhartia[2], Glen Jaross[2], Robert Loughman[3], Natalya Kramarova[1], Zhong Chen[1],
Ghassan Taha[4], Grace Chen[1], and Philippe Xu[5]
[1] Science Systems and Applications, Inc. (SSAI), 10210 Greenbelt Road, Suite 600, Lanham,
Maryland 20706  USA
[2] NASA Goddard Space Flight Center, Greenbelt, Maryland, USA
[3] Hampton University, Hampton, Virginia  USA
[4] GESTAR, Columbia, MD USA
[5] Science Applications International Corporation (SAIC), Lanham, MD
Correspondence to: Leslie Moy (leslie.moy@ssaihq.com)

















## Abstract

One of the largest constraints to the retrieval of accurate ozone profiles from UV backscatter limb sounding sensors is altitude registration. Two methods, the Rayleigh Scattering Attitude Sensing (RSAS) and Absolute Radiance Residual Method (ARRM), have been developed to determine the altitude registration to the accuracy necessary for long-term ozone monitoring. The methods compare model calculations of radiances to measured radiances, and are independent of onboard tracking devices. RSAS determines absolute altitude errors, but because the method is susceptible to aerosol interference, it is limited to latitudes and time periods with minimal aerosols. ARRM can be applied across all seasons and altitudes. However, it is only appropriate for relative altitude error estimates. The application of these methods to Ozone Mapping and Profiler Suite (OMPS) Limb Profiler (LP) measurements showed that, at launch, the OMPS LP instrument had a 1-2 km altitude registration error, resulting in a 50% error in the derived ozone density at some altitudes. Though some of the error has been attributed to thermal shifts in the focal plane of the instrument, most of it appears to be due to misalignment of the spacecraft star trackers or the OMPS LP focal plane with respect to the spacecraft axes. In addition, there are ±200 m seasonally varying errors that could either be due to errors in the spacecraft pointing information or in the geopotential height (GPH) data that we use in our analysis.

**Keywords: altitude registration, OMPS Limb Profiler, RSAS, ARRM, ozone profile, backscattered ultraviolet**



## 1    Introduction

Instruments that measure the solar radiation scattered by the earth's atmosphere in the limb direction provide a low cost way of measuring stratospheric ozone and aerosols from satellites. The technique provides daily full coverage of the sunlit earth from commonly used polar sun-synchronous satellites. To meet the science requirements for monitoring ozone requires the altitude registration of the radiances to be accurate to within ~100 m. For a sensor orbiting at 800 km, this translates into ~6 arcsec accuracy in the pointing direction of the instrument line-of-sight (LOS) with respect to earth's horizon. This is often a difficult if not impossible goal to achieve.

In this paper we critically examine the performance of two methods of altitude registration that compare the radiances measured by the instrument to model calculations of radiances. We discuss the methods' inherent strengths and limitations and then assess their performance using data from the OMPS Limb Profiler (LP), launched onboard the Suomi NPP (SNPP) satellite on October 28, 2011.

One of these techniques, known as Rayleigh Scattering Attitude Sensing (RSAS), is relatively insensitive to instrument radiometric errors and drift. However, it works best when the effect of aerosols on 350 nm/20 km limb radiances are small. Under these conditions, the accuracy of the method is determined by the accuracy of the geopotential height (GPH) data near 3 hPa (~40 km) that is used to calculate the limb radiances. Since aerosol contamination limits the range of latitudes and seasons where RSAS can be applied, we developed the Absolute Radiance Residual Method (ARRM). Although ARRM can be applied more broadly than RSAS, it is more suitable to analyzing relative rather than absolute errors.

We describe the theoretical basis of these two techniques in Section 2, and move on to results for the OMPS LP instrument in Section 3. Finally, we present several validations of our uncertainty estimates in Section 4.

## 2    Theoretical Basis

Most scene-based altitude registration methods applied to limb-scattering instruments take advantage of the fact that the atmospheric Rayleigh scattering measured by these instruments varies by 12-14%/km in the absence of particulate scattering from aerosols and clouds and absorption by trace gases. For wavelengths longer than 310 nm, the limb-scattered radiance has a



significant contribution from diffuse upwelling radiance (DUR), which is affected by
tropospheric clouds, aerosols and surfaces from inside a circular cone whose base extends
hundreds of km to the horizon. At non-ozone absorbing wavelengths DUR can be as much as
half of the measured radiance. Since DUR is difficult to model accurately, all successful altitude
registration methods must be relatively insensitive to variations in it.
The RSAS method, described in Sect. 2.1, employs signal ratios in which the DUR effects
largely cancel. The ARRM, described in Sect. 2.2, uses 295 nm radiances for which ozone
absorption screens the DUR signal. The Knee method, described in Sect. 2.3, has been used
extensively by others (Sioris et al., 2003; Kaiser et al., 2004; Rault et al., 2005; von Savigny et
al., 2005, Taha et al., 2008), but our analysis indicates that it has no advantages over RSAS and
ARRM.

**2.1    Rayleigh Scattering Attitude Sensor (RSAS)**
This technique is named after a sensor that was flown on the Space Shuttle STS-72 in January
1996 (Janz et al., 1996) to test the concept originally proposed by one of the authors (Bhartia) ca
1992. The technique takes advantage of the fact that change in the log of the limb-scattered (LS)
radiance I with altitude z, dlnI/dz, changes by a factor of 3 between 40 km and 20 km for
wavelengths near 350 nm (Fig. 1). This is caused by the exponentially increasing attenuation of
Rayleigh scattering with pressure. At 40 km this attenuation is small and dlnI/dz is largely
determined by dlnP/dz, where P is the atmospheric pressure at altitude z. However, at 20 km the
extinction and scattering nearly cancel where the line of sight (LOS) intersects the Earth radius
vector at a right angle, called the tangent point (TP). Therefore the radiances at 20 km are
relatively insensitive to the exact altitude of the TP. Though several variations of the RSAS
technique have been developed (McPeters et al., 2000; Rault et al., 2005; Taha et al., 2008), we
find that the simplest formulation described below works as well as any other.
If r is the ratio of radiances for wavelength $\lambda$ at altitudes $z_1$ and $z_2$, and $s_1$ and $s_2$ are the vertical
slopes dlnI/dz at those altitudes, then the error in tangent height (TH) can be calculated as
follows:
$$\Delta z = - \frac{\ln(r)_m - \ln(r)_c}{s(z_1) - s(z_2)}$$   (1)
where the subscript m refers to the measured radiance ratios, and c to the ratio calculated using a
radiative transfer model. To get the most accurate estimate of TH error the denominator should





be as large as possible and the uncertainties in estimating the numerator should be small. The
smallest uncertainties in the numerator occur at wavelengths near 350 nm, where trace gas
absorption and aerosol scattering effects are small. Setting z1 to be near 40 km and z2 to be at or
below 20 km maximizes the value of the denominator, typically near 0.10/km. So an accuracy of
0.01 (equal to 1% in radiance ratios) is needed to estimate TH within 100 m.
As Fig. 2 shows, the largest source of noise in estimating the numerator comes from the
mismatch in cloud sensitivity of radiances at the two altitudes. However, this noise is random
and can be reduced by averaging data from multiple orbits.
Aerosols in the instrument's LOS are a more important source of error. Though the effect of
aerosols near 350 nm is small compared to longer wavelengths, it is quite complicated (Fig. 3)
and difficult to model since it is determined by subtle differences between two large effects: the
reduction of Rayleigh scattering by aerosol extinction and the enhancement of limb radiances by
aerosol scattering. In addition, 350 nm LS radiances at 20 km are significantly affected by
variation of aerosols along the LOS because of large Rayleigh attenuation; aerosols in the LOS
close to the sensor contribute more heavily to the radiance than those far away. Though this
effect is similar to the cloud effect mentioned earlier, it is not random because aerosols tend to
have systematic latitudinal variability. Given this complexity, the RSAS method works best in
latitudes and months where the 350 nm aerosol extinction at 20 km is relatively small.
Another potential source of uncertainty in applying the RSAS technique comes from uncertainty
in estimating $r_c$ at 40 km; one needs to have accurate pressure profiles at and above 40 km. If the
pressure profiles are obtained from geopotential height (GPH) profiles provided by
meteorological data assimilation systems, a one-to-one relationship exists between the two errors:
a 100 m error in GPH at 3 hPa translates into ~100 m error in determining TH altitude.

**2.2    Absolute Radiance Residual Method (ARRM)**
This method uses radiances measured by a limb instrument near 295 nm at ~65 km to determine
altitude error. The main advantage of ARRM is that it reduces aerosol contamination effects
because, with the exception of polar mesospheric clouds (PMCs), the atmosphere is typically
free of particulate matter at 65 km. PMCs, which form in the polar summer and are typically
located at 80 km, can significantly affect 65 km limb radiances if they are in the LOS of the
instrument. Fortunately most of the PMC contamination is screened using a 353 nm channel





radiance residual flag at 65 km. Though 295 nm radiances are very ozone sensitive, this
sensitivity drops to less than 0.2% for a 10% change in ozone above 65 km. This sensitivity can
be accounted for by using climatological ozone profiles.
The principal difficulty in applying ARRM comes from the fact that one cannot get accurate
GPH data near 0.1 hPa needed to calculate 295 nm radiances at 65 km. To reduce this error we
developed a variation of a technique that has been used for many years to derive mesospheric
temperature profiles from the vertical slope of Rayleigh-scattered radiances measured by ground-
based UV lidars (McGee et al., 1991). Though it may be possible to derive temperature profiles
using 350 nm limb radiances, we are more interested in using the vertical slope of Rayleigh-
scattered radiances to correct the 295 nm radiance residuals calculated using meteorological data.
The residual at wavelength $\lambda$ at altitude z, defined as $d(\lambda,z)= \ln I_m(\lambda,z)-\ln I_c(\lambda,z)$, is corrected
using 350 nm residuals:
$d_{corr}(\lambda,z) = d(\lambda,z) - [d(350,z) - d(350,z_0)]$                    (2)
where $z_0$ is a normalization altitude.
The 350 nm differential residuals on the right side provide an estimate of the relative error in
calculating radiances using meteorological data between z and $z_0$. Since this error should be
wavelength independent, we can use this term to correct the residuals at any wavelength,
assuming that the meteorological data at $z_0$ is accurate and that the 350 nm wavelength is well
calibrated. The large response of OMPS LP at 350 nm results in signals that are the least affected
by out-of-band stray light.
The TH error estimated using this method is given by:
$\Delta z = \dfrac{d_{corr}(\lambda,z)}{s(\lambda,z)-[s(350,z)-s(350,z_0)]}$                    (3)
We are minimizing ozone profile sensitivity by applying this method to radiances at wavelengths
shorter than 300 nm. At longer wavelengths DUR makes the LS radiances sensitive to total
column ozone at all altitudes. At 295 nm, the use of z near 65 km provides low ozone sensitivity.
Though it is best to set $z_0$ as low as possible to minimize GPH caused errors, aerosol
contamination limits the value to around 40 km.
ARRM has two primary uncertainties. Since 1% error in radiance calibration produces ~70 m
error in determining the TH, this method requires accurate radiances (or sun-normalized
radiances) and may be affected by instrument degradation. Though the absolute accuracy of





ARRM may not be as good as RSAS, this method can be applied at latitudes/seasons where
RSAS cannot be applied reliably because of aerosol contamination. And like RSAS, this method
is also sensitive to errors in GPH profile near 3 hPa, which are used for calculating 350 nm
radiances at 40 km.

**2.3**    **"Knee Method"**
The name of this method is derived from the characteristic knee shape of the limb radiance
profiles (Fig. 4). Above the knee the radiances decrease with altitude due to exponential decrease
in Rayleigh scattering and ozone density. Below the knee the ozone absorption becomes so large
that it essentially blocks most of the Rayleigh-scattered radiation from reaching the satellite,
making the radiances insensitive to atmospheric pressure. This characteristic shape allows one to
estimate altitude registration error in a manner very similar to that of RSAS. The principal
advantage of this method is that one can use shorter wavelengths where aerosols are not a
problem. However, this comes at a penalty; the method requires accurate ozone and pressure
profiles near and above the knee region. Radiative transfer calculations using climatological
ozone profiles indicate that a 10% error in assumed ozone density (at all altitudes) will produce
about a 250 m error in altitude registration (Fig. 5). The method also has a sensitivity to GPH
errors that is similar to RSAS and ARRM. In our view this method provides no compelling
advantage over comparing the ozone profiles retrieved from a limb scattering sensor with other
ozone sensors to determine altitude registration errors. Indeed, such direct ozone comparisons are
simpler and more reliable if the altitude registration error is the largest error source, and we use
this technique to evaluate the results of RSAS and ARRM in Sect. 3.

**3**    **Results**
In this section we discuss altitude registration errors in OMPS LP radiances determined first by
"Slit Edge" analysis of the instrument focal plane image and then by the application of the RSAS
technique. The remaining errors are analyzed using ARRM.

**3.1**    **Slit Edge Results**
The OMPS LP sensor utilizes a two-dimensional charge coupled device (CCD) detector to
capture spectrally dispersed (along the 740 pixel row dimension) and vertically distributed (along



the 340 pixel column dimension) radiation (Fig. 6). Three long vertical entrance slits spaced 4.25°

apart produce three distinct images of the atmosphere that are collected simultaneously on the

single CCD. The resulting limb radiance profile from the center slit is aligned very closely to the

satellite ground track with tangent points trailing approximately 3000 km south of the sub-

satellite point. The east and west slit images are separated in longitude by 2.25° (250 km at their

tangent points) from that of the center slit.

An unexpected thermal sensitivity was discovered in the LP instrument soon after launch (Jaross

et al., 2014). Expansion of the LP instrument's entrance baffle as the sun illuminates it midway

through the northern hemisphere causes mirrors in the telescope to rotate slightly, which in turn

moves the limb radiance image on the detector. Since there are separate mirrors for each entrance

slit, the three slit images move independently. These image motions cause misregistration of

both the vertical pointing and center wavelength of each pixel. Vertical pointing changes are

detected most clearly by observing the location (detector column) of the lower slit edge, which

has a sharp signal gradient. Figure 7 contains plots of the average edge locations in the vertical

(altitude) dimension along the orbit. These pointing shifts are very repeatable (ranging only ±15

m at a given point in the orbit over a year).

Since the same slit edge analysis can be applied to pre-launch test data, it is possible to obtain the

pixel line of sight shift relative to its calibrated value in the spacecraft reference frame. There is

no evidence of image distortion so this shift is the same for all detector pixels within a slit image.

The edge analysis indicates the three slit edges shifted by the equivalent of 570/470/950 m

(east/center/west slits, respectively) at the middle of an orbit relative to pre-launch measurements.

A mean sensor temperature decrease exceeding 25° C from ground to on-orbit conditions is the

suspected cause. We believe there are no additional uncorrected pointing shifts arising from

within the LP instrument. An error or change in the alignment of the instrument with respect to

S/C axes is not detectable using this method.

**Section 3.2   RSAS results**

We use the radiative transfer code described by Loughman et al. (2015) to estimate 350 nm

radiances. Since the 40/20 km radiance ratio is not sensitive to polarization effects, we use the

faster scalar code rather than the full vector one to calculate DUR. The calculations are done

assuming a pure Rayleigh atmosphere bounded by a Lambertian reflecting surface at 1013.25





hPa. The reflectivity of this surface is calculated using limb measurements at 40 km. However,
both measurements and calculations show that the ratio of 40/20 km radiances is not affected by
reflectivity or surface pressure and there is no discernible cloud effect. Since $NO_2$ only has a
very small (<0.5%) effect on 350 nm radiances, climatological $NO_2$ profiles are used in the
calculation. We use OMPS LP retrieved ozone profiles. The RSAS analysis is not sensitive to
ozone assumptions because it uses the ozone insensitive 350 nm radiances.
We estimate pressure and temperature versus altitude at the LP measurement locations and time
from the Modern-Era Retrospective Analysis for Research and Application (MERRA) data
(GEOS-5 FP_IT Np) from the Global Modeling and Assimilation Office (GMAO) at NASA
Goddard Space Flight Center (GSFC). The data are provided as geopotential heights (GPH) at 42
pressures from the surface to 0.1 hPa, on a 0.5° latitude x 0.625° longitude horizontal resolution
grid, and at a 3 hour interval. The GPH is converted to geometric height using a standard formula
that takes into account the variation of gravity with latitude and elevation.
As discussed in Sect. 2.1, RSAS results are affected by aerosols near 20 km. Aerosol profiles
derived from the Optical Spectrograph and InfraRed Imaging System (OSIRIS) data (Llewellyn
et al, 2004; Bourassa et al, 2007) indicate that tropical aerosols reached a minimum value (during
the OMPS lifetime) just before the eruption of the Kelud volcano in Indonesia on February 14,
2014 (Fig. 8). We have therefore chosen to use equatorial RSAS data before the eruption to
estimate the altitude registration errors (listed in Table 1). These TH errors range between ~1 and
~1.5 km for the three slits, and have been applied to the Version 2 OMPS LP Ozone data set.
Radiative transfer calculations using OSIRIS-derived aerosol profiles indicate that the aerosol
caused errors in the results shown are less than 100 m.

**Section 3.3     ARRM results**
We utilized the same radiative transfer code and profile inputs used for RSAS to calculate
radiances at 295 nm. Ozone concentrations from the OMPS LP retrievals were used. As
mentioned previously, though 295 nm radiances are very ozone sensitive, this sensitivity drops to
less than 0.2% for a 10% change in ozone above 65 km. As discussed in Sect. 2.2, the absolute
accuracy of ARRM may not be as good as RSAS, but this method can be applied at latitudes and
seasons where RSAS cannot be applied reliably because of aerosol contamination.



Time dependent plots (Fig. 9) show negative pointing trends of approximately 100 m over the
four years of data, and even larger seasonal variations depending on the latitude band and slit
over the four years of data. Much of this trend is the result of a 6 arcsec (a TH change of ~100m)
spacecraft pitch adjustment that occurred on 25 April 2013. Figure 9 clearly shows this abrupt
change.
The largest disagreement between the 3 slits (~400 m) occurs in the high northern hemisphere. In
the southern hemisphere the disagreement is closer to 100 m. In addition we see ±200 m
seasonally varying errors that could be due to either true pointing changes or errors in the GPH
data that we used in our analysis. Such variations in 295 nm radiances cannot be explained by
known seasonality in ozone concentration at 65 km.
The ARRM method is designed to accommodate stray light errors that are independent of
wavelength. No additional TH errors occur when stray light at 65km is the same at 295 and
350nm. The ground characterization of Limb sensor stray light indicates a small wavelength
dependence (Jaross, 2014), and this is removed in ground processing. Our subsequent
comparisons with RTM predictions indicate that residual stray light errors at 65 km have a daily
mean bias that translates to less than 100 meters in TH.
The ARRM analysis shows a distinct latitude dependence (Fig. 10) with some seasonal
differences. While it is tempting to attribute this entirely to TH error, we conservatively do not
apply the ARRM results to our data since the uncertainties are of the same magnitude. As with
RSAS, this method is sensitive to errors in GPH profile near 3 hPa used for calculating 350 nm
radiances at 40 km (see Sect. 4.1). Further analysis is needed to determine the precise cause of
the remaining errors.

**Section 4        Validation**
In this section we consider uncertainties in the parameters used to derive TH errors to indirectly
validate the results shown in Sect. 3, and to estimate the remaining uncertainties in the LP TH
due to errors in these parameters.   Section 4.1 focuses on the validation of 3 hPa GPH
information  from MERRA  that was assumed as the truth in our calculations.  In Section 4.2 we
consider the reflectivity measured by the OMPS nadir sensor to validate LP-measured radiances
at 350 nm, which vary by ~14%/km near 40 km. Finally, in Sect. 4.3 we compare the LP-derived





ozone mixing ratio at 3 hPa with the Microwave Limb Sounder (MLS). For the validation studies
in this section, the OMPS LP TH has been corrected with the errors listed in Table 1.

### Section 4.1      GPH comparison

Errors in the GPH profile assumptions directly translate into TH errors. Although the 3hPA GPH
varies over 4 km along an orbit, a comparison of daily averaged values from MLS and MERRA
show differences that are usually less than 200 m (Fig. 11). These differences do not directly
explain the latitude dependence of TH errors shown in Fig. 10, but do provide an estimate of the
magnitude of errors caused by the use of MERRA GPH in our radiative transfer calculations.
Better agreement seen at the poles may simply be due to the fact that there are not many
measurements at these latitudes and both may be influenced by the same climatology. As a result,
it is not clear how these GPH errors influence the ARRM results. However, in Section 4.3 we
discuss some suggestive but inconclusive results untangling GPH errors from TH errors.

### Section 4.2      Radiances comparison

We previously described (in Section 2) the difficulty modeling DUR caused by scene
heterogeneity and aerosols. Both the RSAS method and ARRM depend upon an accurate model
for DUR at 40 km relative to other altitudes, and any model errors translate directly (Equations 1
and 3) into false estimates in the TH errors.
We estimate the DUR modeling error by comparing LP measured and modeled 353 nm radiances
at 3 hPa. The radiances are modeled using an independent, nearly simultaneous measure of
reflectivity from the surface-atmosphere system derived from the OMPS Nadir instrument at 340
nm. The reflectivity derived from the 50x50 km nadir-view measurements are relatively
insensitive to DUR effects (compared to reflectivity derived from the LP measurements). With
better reflectivity assumptions the model/measurement comparisons offer a lower bound of the
effect of DUR modeling errors.
The radiance comparison, shown in Fig. 11, suggests model or calibration errors of 2-3% on
average, plus structures caused by the limb and nadir scene mismatch. If this error were
attributed solely to the limb modeled DUR effect, the resulting TH error would be less than +/-
200 m. There is no evidence of either a seasonal or a latitude dependence in the four days of



comparisons, meaning that DUR effects cannot explain the robust seasonal and latitudinal
variations seen in ARRM results (Fig. 9 and Fig. 10). These model/measurement comparisons
provide an estimate of errors related to incomplete modeling of DUR and inhomogeneous
surface albedo included in our RSAS and ARRM results. We therefore conclude those variations
arise from errors in the GPH scale or from true TH variations.

**Section 4.3    Ozone comparison**
At 3 hPa limb ozone retrievals are very sensitive to TH errors, with 20 to 25% per km change in
ozone concentration (see Fig. 5). Similar to the Knee Method, we can use this sensitivity to
gauge the residual TH errors. We compare LP ozone retrievals against Aura MLS v4 ozone
retrievals at 3 hPa (near 40 km) (Fig. 13). While the latitudinal patterns of differences
significantly vary with season, we find agreement within ±10% over all seasons and latitude
bands. If completely interpreted as due to TH error, a 10% difference would translate to less than
500 m error. These comparisons confirm a residual uncertainty in our scene-based altitude
registration techniques of ±200m.
The ARRM method has displayed the ability to track any drifts or sudden changes of 50 m (Sect.
3.3), and time series of TH error derived from the ARRM method  track very closely to the time
series of the LP/MLS 3 hPa ozone differences (Fig. 14). The highest correlation (0.76) was found
at 45° south latitude, with considerable smaller values in the northern hemisphere (0.30 at 60°
north).  Whether this suggests the ARRM results can be attributed solely to TH errors has not
been determined yet.
Both ARRM and LP/MLS ozone comparison depend upon accurate TH and MERRA
information, and in the same way.  So, while these results suggest some confidence in the
ARRM technique, we cannot assign the correlation shown in Fig. 14 to only a TH error or a
MERRA error. It is important to note that MLS ozone profiles are reported as volume mixing
ratio on a vertical pressure grid, while the LP algorithm retrieves ozone as number density on an
altitude grid. Thus, in order to compare LP and MLS ozone retrievals we had to convert ozone
units using MERRA temperature and GPH profiles. This conversion inevitably introduces errors
in MERRA GPH into the ozone comparisons. Therefore ozone differences between LP and MLS
ozone retrievals not only depend on the LP TH error, but on errors in MERRA GPH as well as
on errors in the retrieval algorithms and instrumental sampling (geophysical noise).  Furthermore,



analysis of LP and MLS ozone retrievals indicates a large daily ozone variability within a 5-degree latitude bin at 3 hPa that ranges from 2% in the tropics to 20% at high latitudes with the seasonal maximum during austral winters (results are not shown here), which can give readers a sense of geophysical ozone variability. In consideration of all of the above factors, we remain cautious in making definite conclusions and applying time-dependent corrections for the LP TH at this time; further analysis and comparisons with independent ozone observations (like SAGE III) are needed to confirm the results.

**Section 5      Conclusions**

Accurate altitude registration is key to the success of the limb scattering measurement technique. We have described two scene-based techniques that together provide highly precise and accurate estimates of the tangent height. These altitude registration techniques are inexpensive and more comprehensive than external sources of attitude information, such as star trackers mounted on the spacecraft. Though star trackers are highly accurate devices, translation of that accuracy to the limb scene is also not without uncertainty, as we have seen with the SNPP spacecraft. Though we were able to calibrate thermal sensitivities within the OMPS instrument, we have yet to identify the source of 1-1.5 km pointing errors derived from RSAS (see Table 1). These may arise from mounting  offsets  of the instrument and star trackers, or from spacecraft flexure between the two.

The RSAS and ARRM techniques are complementary because the former is accurate to ±200 m, but only under limited conditions. The accuracy of ARRM cannot be easily established, but it has a precision also within ±200 m. We believe this results in small, less than 100 m, trend uncertainties for sufficiently long time series, as demonstrated by the OMPS ARRM record.

The single largest source of uncertainty in both techniques is knowledge of the atmospheric pressure vertical profile, which must be provided from external sources. Given uncertainties in GPH data, as seen in the MLS comparison, as both well as uncertainties in our ozone retrieval algorithm (not related to TH error), it is currently not possible to tell if the latitudinal and seasonal variations seen in ARRM results are caused by TH error. Further work will be needed to understand their cause. We have shown, however, that ARRM is capable of multi-year trend



uncertainties that are on the order of 100m or smaller. Furthermore, the two TH registration methods discussed in the paper allow us to track any drifts or sudden changes in our altitude registration to better than 50 m, which is the minimum level necessary to derive accurate ozone trends from a limb technique.

Acknowledgements: The authors gratefully acknowledge the assistance of NASA's Limb Processing Team in providing the data used in this paper. We would also like to thank Dave Flittner, Ernest Nyaku and Didier Rault, helped with the development and updates of the RT model. Finally, we'd like to acknowledge the role Didier played in laying the groundwork for the OMPS limb retrieval algorithm

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



Table 1: RSAS results at the equator before the Kelud eruption 2014 February. The time period
had a minimum value (during OMPS life time) and was chosen using OSIRIS measurements
(Fig.8).

| TH error, km | EAST | CENTER | WEST |
|---|---|---|---|
| RSAS results | 1.12 | 1.37 | 1.52 |

