# Peer review of "Altitude Registration of Limb-Scattered Radiation"

_Atmospheric Measurement Techniques, 2016_

## Author Comment (AC1) · 8 Jun 2016

[revised manuscript text omitted]

**Figure 1:** Figure a shows calculated 350 nm radiances as a function of altitude, normalized to 40.5km. The calculation models the OMPS LP field of view and includes no aerosols. The shape of the curve is caused by competition between molecular scattering, which increases roughly linearly with pressure, and attenuation which becomes important when the Rayleigh optical thicknesses near the tangent point starts to become large. Attenuation causes the slope of 350 nm radiances to change sharply between 40 and 20 km (Fig. 1b). Since the ratio of 40 to 20 km radiances at 350 nm varies by 8-10%/km, one can estimate altitude registration errors by comparing measured ratios with ratios calculated using meteorological data.

[Figure]

**Figure 2:** The 353 nm sun-normalized radiances from one orbit of OMPS LP (central slit) taken on Feb. 2, 2012. The blue line shows 40.5 km values and the green line shows 20.5 km values (divided by 8 to put both curves on a similar scale) versus latitude. Since the global aerosol loading on this day was small, the short scale features in both curves are largely caused by variations in cloud and surface albedo. The 20.5 km curve has sharper features and appears to be shifted towards toward the South Pole. This is because large Rayleigh attenuation at 20.5 km causes the radiances to have much higher sensitivity to the atmosphere on the satellite side of the tangent point (TP), while 40.5 km radiances have similar sensitivities to both sides. This effect creates large noise in applying the RSAS technique to orbital data. However, since the noise varies randomly from orbit to orbit, it can be reduced by averaging data from multiple orbits. Figure 6 of Loughman et al. (2015) is an example of how the contributions become asymmetric about the tangent point at lower THs.

[Figure]

**Figure 3:** The ratio of 353 nm limb-scattered radiances at 20.5 km with and without aerosols as a function of latitude. A nominal latitude-independent aerosol extinction profile was used in the calculation for the OMPS LP viewing geometry on Feb 2, 2012. The strong latitude dependence is caused by an order of magnitude change in aerosol scattering phase function with latitude combined with the attenuation of Rayleigh-scattered radiation by aerosols along the line-of-sight (LOS). In the southern hemisphere, where LP measures aerosols in the backscatter direction, the latter effect dominates and the radiation decreases. The net effect is very sensitive to altitude, variation of aerosol extinction profile along the LOS, and aerosol particle size distribution, and is therefore difficult to calculate accurately.

[Figure]

**Figure 4:** Figure a shows calculated 305 nm radiances assuming no aerosols as a function of altitude. The slope (Fig. 4b) is caused by competition between Rayleigh scattering and ozone absorption near the altitude of maximum radiance, ~44 km. Above 55 km the sensitivity is nearly constant in height, ~13%/km at 65 km. Above the knee the radiances decrease with altitude due to the exponential decrease in Rayleigh scattering and ozone density. Below the knee the ozone absorption becomes so large that it essentially blocks most of the Rayleigh-scattered radiation from reaching the satellite, making the radiances insensitive to atmospheric pressure. This characteristic knee shape allows one to estimate altitude registration error in a manner very similar to that of RSAS, but also makes it is very susceptible to ozone profile assumptions, as illustrated in Fig. 5.

[Figure]

**Figure 5:** Typical ozone profile in the tropics (left panel) which peaks between 25 and 30 km. By shifting the ozone profile we can estimate an order and pattern of error in ozone profiles due to TH shift (right panel). Errors in ozone retrievals are within 8% at 40 km from TH errors of 300 m. Errors are least sensitive at the ozone peak, and are more variable below.

[Figure]

**Figure 6:** OMPS LP CCD high gain earth viewing radiance images for the 3 slits. The wavelength range for each image is 270 to 1050 nm and the minimal altitude range is 0 to 80 km. The CCD has 740 pixels in the wavelength dimension. There are 340 pixels in the spatial dimension; the high gain images occupy the lower half of the CCD (pixels 0 to 170). The spatial extent of each slit's image on the detector is limited by the vertical length of that slit. The lower slit edges (nearest the Earth surface) provide a high contrast signal cutoff that can be monitored for movement.

[Figure]

**Figure 7:** Slit edge results for the three slits (Green=East Slit, Red=Center Slit, Blue=West Slit) plotted against time since Southern Terminator crossing. A 1 pixel shift corresponds to a 965 m TH shift. The offsets are stable from the southern terminator to the mid latitude northern hemisphere where the exposure to the sun increases thermal effects.

[Figure]

**Figure 8:** Time series of OSIRIS aerosol extinction profiles above the tropopause (dashed line). The large concentration in 2012 are due to the June 2011 Nabro eruption in Eritrea. The aerosols at 20 km reached a minimum value (during OMPS life time) just before the eruption of the Kelud Volcano on 14 February 2014.

[Figure]

**Figure 9:** Time dependent plots of TH errors from ARRM analysis at 5 latitude bands for the 3 slits. Values are normalized at the Equator just prior to the Kelud eruption on February 14, 2014 based on the RSAS results summarized in Table 1. Arrows indicate a 12 arcsec pointing adjustment to one of the two spacecraft star trackers on April 25, 2013. The resulting 100 m TH shift can be seen most clearly by comparing 2012 and 2013 results. Slit discrepancies and seasonal dependencies of +/-200 m can also be seen.

[Figure]

**Figure 10:** Average (over the ~4 year study period) ARRM results by latitude and seasons (MAM-green, JJA-red, SON-purple, DJF-blue) for Center Slit. On average, excluding the extreme polar regions there appears to be a 300 m TH change over an orbit.

[Figure]

**Figure 11:** Daily 5 degree zonal means of GPH from MLS (blue), GMAO (green), and the difference MLS-GMAO (red) at 3 hPa GPH for four cardinal days. Note that despite a 2 to 4 km change over an orbit, the differences are generally within 200 m. These differences provide an estimate of the errors caused by the use of MERRA GPH in our radiative transfer calculations. Better agreement seen at the poles may simply be due to the fact that there are not many measurements at these latitudes and both may be influenced by the same climatology.

[Figure]

**Figure 12:** We have estimated the DUR modeling error by comparing 353 nm measured and modeled radiances at 3 hPa. The radiances are modeled using an independent, nearly simultaneous measure of surface reflectivity derived from the OMPS Nadir instrument at 340 nm. The 50x50 km nadir-view measurements are relatively insensitive to DUR effects. The radiance differences (given for the same four cardinal days as in Fig. 10) suggests model or calibration errors of 2-3% on average, plus structure caused by the contributing limb and nadir scene mismatch. If this error were attributed solely to the limb model and only at one altitude, the resulting TH error would be less than +/-200 m. There is no evidence of either a seasonal or a latitude dependence in the comparison, meaning that DUR effects cannot explain the variations seen in Fig. 9.

[Figure]

**Figure 13:** Daily 5 degree zonal means of ozone from MLS (blue), GMAO (green), and the MLS- GMAO differences (red) at 3 hPa GPH for four cardinal days. The differences are generally within 6% which if completely attributed to TH error would be ~200 m.

[Figure]

**Figure 14:** The time series of daily zonal mean ozone differences (%) between OMPS LP and Aura MLS for the 3 slits (top). OMPS LP profiles are corrected with the TH error shown in Table 1. Note the similarity in time-dependent patterns for ozone differences and TH error derived from AARM method (bottom, reprinted from Fig.9). The fact that these two completely independent methods show very similar patterns give us additional confidence in the AARM method.

---

## Referee Comment (RC1) · Anonymous Referee #2 · 15 Jun 2016

**1  Summary**

This work describes different methods to determine the pointing error of satellite measurements in limb geometry. The accurate knowledge of this error is required for precise information on stratospheric trace gas and aerosol extinction profiles. Rayleigh Scattering Attitude Sensing (RSAS), Absolute Radiance Residual Method (ARRM) and the Knee method are evaluated using measurements of the Ozone Mapping and Profile Suite (OMPS) Limb Profile. All three methods are categorized into different latitude regions where they are applicable depending on the stratospheric aerosol distribution. The paper does comprehensively describe these methods and lists potential difficulties that can be investigated in future studies. However, the paper would benefit from some corrections and minor revisions. In the following I will list general suggestions, followed by suggested corrections to grammar and spelling and comments on the figures.

[Figure]

**2 Minor Revisions/Suggestions**

- Line 64 to 67: Maybe focus on profile information instead of low cost as satellite measurements are usually very expensive.

- Line 68: Could you further explain, what are these science requirements that result in a TH accuracy of 100 m.

- Line 79: "effect of aerosols [...] are small" - small compared to what? How low does the aerosol extinction have to be?

- Line 112 to 114: What about particles (like meteoric dust) at altitudes $\geq$ 35 km ?

- Line 119 to 121: What is the effect of the vertical resolution? Typically, limb measurements can only provide limited information on dlnI/dz.

- Line 144: "... estimate $r_c$..." Why do you estimate a simulated value? Maybe write "simulate" instead?

- Line 152/153 and Line 182: Again the question of the influence of meteoric dust arises. If the extinction of those particles is too small to be of importance, than I would suggest to at least mention this.

- Line 158: Where do you get the climatological ozone values from? And how do you deal with the transition between measured profile and climatology?

- Line 207: You say, that the Knee method is unreliable and depends on the O3 profile - how can you use it to evaluate the other two methods?

- Line 260: Have you thought of using ECMWF Operational Analysis data? It is not well suited for long term analysis, since it changes model configuration frequently. However, in order to quantify the GPH uncertainties you can use it for

case studies: Since February 2006 this model reaches up to 0.01 hPa (80km) with a temporal resolution of 6h (should be sufficient for the higher atmosphere).

- Line 265: How do you deal with high altitude clouds for equatorial RSAS data?

- Line 360: You mention MERRA GPH, but not the temperature. What about uncertainties in the MERRA temperatures, mentioned in Line 358?

**3  Grammar/Spelling/Typos/Suggestions**

The suggestions listed below are according to my best knowledge. Not all items are mandatory corrections.

- Line 70: "earth's" → "Earths" (capital E and no use of ')

- Line 70: "difficult if not impossible" - this phrase is reoccurring. I suggest to reformulate it.

- Line 73: "... that compare the radiances measured by the instrument to model calculations of radiances." → "... that compare measured and simulated radiances."

- Line 74: "methods'" → "methods" (no ')

- Line 79: "... radiances are small." compared to what? I would also suggest to reformulate the following sentence as it appears more complicated than necessary.

- Line 84: "... than absolute errors" → Add "in limb altitude registration."

- Line 92: "from aerosols" → "by aerosols"
- Line 95: "tropospheric clouds, aerosols and surfaces" → "tropospheric clouds, aerosols and surface albedos". I think it would be even better to separate clouds and aerosol that are within the "circular cone" and clouds and surface albedo that are below said cone in its footprint.

- Line 97: "difficult if not impossible" - this formulation is reoccurring - maybe reformulate it.

- Line 98: "variations in it." → "variations within."

- Line 108: "... by one of the authors (Bhartia) ca 1992." → "... by Bhartia in 1992." I am not used to the type of quotation you chose. Also, "ca" should be "ca.".

- Line 109: "that change... changes" → "that the gradient... changes"

- Line 127/128: "... to be at or below 20 km..." → "... to be ≤ 20 km..." (there is a less-equal sign in word and latex)

- Line 133: "... more important..." → "... more significant..."

- Line 135: "... it is quite complicated (Fig. 3) and difficult to model since it is determined by subtle..." → "... it is difficult to model (Fig. 3) due to determination of subtle..."

- Line 139: "... more heavily..." → "... more..." (leave out heavily)

- Line 144: "... at and above 40 km." → "... for altitudes ≥ 40 km."

- Line 159: "The principal difficulty ... at 65 km." (full sentence) → "The main difficulty in applying ARRM is the inaccuracy of GPH data near 0.1 hPa required to calculate 295 nm radiances at 65 km."

- Line 163: "Though it may be... meteorological data." - I dont understand this sentence. Could you reformulate it?

- Line 170: "... right side provide..." → "... right side of equation (2) provide..."

- Line 194: "... the knee the ozone..." → "... the knee ozone..." (you can cancel the "the" in front of ozone)

- Line 196: "... shape allows one to..." → "... shape allows to..."

- Line 197: "The principal advantage of this method is that one can use shorter wavelengths where aerosols are not a problem." → "As one advantage of this method shorter wavelengths with less sensitivity to aerosol can be used."

- Line 199: "However, this comes at a penalty; the method..." → "However, the method..."

- Line 203: "... errors that is..." → "... errors that are..."

- Line 251: "... are used in the..." → "... are sufficient for the..."

- Line 284: "... is closer to 100 m." → "... is about 100 m."

- Line 301: "...  14%/km near 40 km." → "...  14 %/km around 40 km."

- Line 306: "3hPA" → "3 hPa"

- Line 312: "...  and both may..." (both what?)  maybe "...  and both values of the GPH may..."

- Line 321: "measure" → "measurements"

- Line 327: "Fig. 11" → I think you mean Fig. 12. If not: Fig 12 is not referenced anywhere else in the text."

- Line 348 to 350: Could you plot this correlation?

- Line 358: "... ozone units..." → "... ozone number densities to mixing ratios..."

- Line 373: "attitude" → "altitude"

- Line 374: "... is also not..." → "... is not..."

- Line 376: "... as we have seen with the SNPP spacecraft." - Do you have sources for this? What are you referring to here?

- Line 382: "... precision also within..." → "... precision within..."

- Line 386/387: "... of the atmospheric pressure vertical profile..." → "... of the vertical profiles of pressure and temperature..."

- Line 388: "... as both well as..." → "... as well as..."

- Line 391: "... that ARRM is capable of multi-year trend..." → "... that ARRM is capable of deriving multi-year trend..."

**4  Figures**

- All axes should have a unit description or (AR) for arbitrary unit.

- All unit descriptions should be uniform. So either choose "(unit)", e.g. in Fig. 4 "Altitude (km)", or ", unit" as in Fig. 1 "Altitude, km"

- All axes should have the same spelling for the label (e.g., Latitude vs. latitude)

- I would suggest to write numbers below 13 as words ("one" instead of "1")

- Figure 1: "normalized to 40.5 km" → "normalized at 40.5 km"

- "field of view and includes no aerosols." → "field of view without aerosols."

- "is caused by" → "originates from the"

- "tangent point starts" → "tangent point start"

- "varies by 8-10%/km" With respect to what? Is it variation for the whole dataset?

- Figure 3: Maybe show the dependency on the scattering angle, perhaps by including an axes for the scattering angle?

- Figure 4: "no aerosols as a function of altitude" Do you mean no aerosols? Or altitude independent aerosol extinction? Please clarify.

- Figure 6: Altitude != TH/Elevation - I think both terms are mixed up in the y axes description. It would also help to include a more detailed figure description on west/center/east slits as the meaning became clear only after reading the main text.

- Figure 8: Is this really the tropopause or just the 380 K isentropic surface?

- Figure 11+12: Figure description is the same as in the main text. I suggest to reformulate the figure description.

---

## Author Comment (AC2) · 22 Jun 2016

We would like to thank the reviewer for their considered comments. Below is our response and attached in a new version of the paper.

Minor Revisions/Suggestions • Line 64 to 67: Maybe focus on profile information instead of low cost as satellite measurements are usually very expensive. Included words to focus on profile information provided by limb sensors. • Line 68: Could you further explain, what are these science requirements that result in a TH accuracy of 100 m. Rewritten: To meet long-term ozone monitoring needs (3% precision between 15 and 50 km) requires the altitude registration of the radiances to be accurate to within ∼100 m. • Line 79: "effect of aerosols [...] are small" - small compared to what? How low does the aerosol extinction have to be? We can never be certain we are aerosol free we can only minimize the aerosol contamination – changed text to explain this. • Line 112 to 114: What about particles (like meteoric dust) at altitudes 35 km ? The RSAS method determines the absolute TH error. We used results before the Kelud eruption which had the smallest values (the cleanest atmosphere) to normalize the ARRM results. The 20 and 40 km altitudes used in the RSAS method are not in the Junge layer. • Line 119 to 121: What is the effect of the vertical resolution? Typically, limb measurements can only provide limited information on dlnI/dz. OMPS LP has a vertical resolution of ∼2km. However, the calculation shown in Figure 1 shows the slope is ∼linear near 20km and ∼vertical near 40km so vertical resolution does not constrain our determination of the slope. • Line 144: "... estimate rc..." Why do you estimate a simulated value? Maybe write "simulate" instead? Done. • Line 152/153 and Line 182: Again the question of the influence of meteoric dust arises. If the extinction of those particles is too small to be of importance, than I would suggest to at least mention this. We do not believe there is persistent dust at 65km, and passing meteoric dust would be averaged out in the zonal means. • Line 158: Where do you get the climatological ozone values from? And how do you deal with the transition between measured profile and climatology? Our climatology is GMAO (GEOS-5 FP_IT Np). Our point is that whatever ozone values we use at 65km, it will not affect our results greatly because of the sensitivity of the 295nm radiance to ozone above 65 km is small. • Line 207: You say, that the Knee method is unreliable and depends on the O3 profile - how can you use it to evaluate the other two methods? We do not use the Knee method for validation. We use ozone comparisons – the sentence is rewritten to make this clearer. • Line 260: Have you thought of using ECMWF Operational Analysis data? It is not well suited for long term analysis, since it changes model configuration frequently. However, in order to quantify the GPH uncertainties you can use it for case studies: Since February 2006 this model reaches up to 0.01 hPa (80km) with a temporal resolution of 6h (should be sufficient for the higher atmosphere). Thank you for this suggestion. We will take this up in future work. • Line 265: How do you deal with high altitude clouds for equatorial RSAS data? We first considered RSAS at the South Pole where the aerosols are normally at a minimum. When comparing it to the equatorial region before the Kelud eruption we found it to be cleaner. So we conclude the contamination was at a minimum then. • Line 360: You mention MERRA GPH, but not the temperature. What about un- certainties in the MERRA temperatures, mentioned in Line 358? Sentence includes temperature now. 3 Grammar/Spelling/Typos/Suggestions The suggestions listed below are according to my best knowledge. Not all items are mandatory corrections. • Line 70: "earth's" !"Earths" (capital E and no use of ') Earth was capitalized but we kept the apostrophe. • Line 70: "difficult if not impossible" - this phrase is reoccurring. I suggest to reformulate it. Done • Line 73: "... that compare the radiances measured by the instrument to model calculations of radiances." ! "... that compare measured and simulated radi- ances." Done • Line 74: "methods'" ! "methods" (no ') Done • Line 79: "... radiances are small." compared to what? I would also suggest to re- formulate the following sentence as it appears more complicated than necessary. Rewritten for clarity. • Line 84: "... than absolute errors" ! Add "in limb altitude registration." Done • Line 92: "from aerosols" ! "by aerosols" Done • Line 95: "tropospheric clouds, aerosols and surfaces" ! "tropospheric clouds, aerosols and surface albedos". I think it would be even better to separate clouds and aerosol that are within the "circular cone" and clouds and surface albedo that are below said cone in its footprint. Done • Line 97: "difficult if not impossible" - this formulation is reoccurring - maybe reformulate it. Done • Line 98: "variations in it." ! "variations within." Done • Line 108: "... by one of the authors (Bhartia) ca 1992." ! "... by Bhartia in 1992." I am not used to the type of quotation you chose. Also, "ca" should be "ca.". Done • Line 109: "that change... changes" ! "that the gradient... changes" Done • Line 127/128: "... to be at or below 20 km..." ! "... to be 20 km..." (there is a less-equal sign in word and latex) Done • Line 133: "... more important..." ! "... more significant..." Done • Line 135: "... it is quite complicated (Fig. 3) and difficult to model since it is determined by subtle..." ! "... it is difficult to model (Fig. 3) due to determination of subtle..." Done • Line 139: "... more heavily..." ! "... more..." (leave out heavily) Done • Line 144: "... at and above 40 km." ! "... for altitudes 40 km." Done • Line 159: "The principal difficulty ... at 65 km." (full sentence) ! "The main difficulty in applying ARRM is the inaccuracy of

GPH data near 0.1 hPa required to calculate 295 nm radiances at 65 km." Done •
Line 163: "Though it may be... meteorological data." - I dont understand this sentence.
Could you reformulate it? Rewritten • Line 170: "... right side provide..." ! "... right
side of equation (2) provide..."Done • Line 194: "... the knee the ozone..." ! "...
the knee ozone..." (you can cancel the "the" in front of ozone) Done • Line 196: "...
shape allows one to..." ! "... shape allows to..." Done • Line 197:"The principal ad-
vantage of this method is that one can use shorter wavelengths where aerosols are not
a problem."!"As one advantage of this method shorter wavelengths with less sensitivity
to aerosol can be used."Rewritten • Line 199: "However, this comes at a penalty;
the method..." ! "However, the method..." Done. • Line 203: "... errors that is..." !
"... errors that are..." Done • Line 251: "... are used in the..." ! "... are sufficient for
the..." Done • Line 284: "... is closer to 100 m." ! "... is about 100 m." Done •
Line 301: "... 14%/km near 40 km." ! "... 14 %/km around 40 km." Done • Line
306: "3hPA" ! "3 hPa" Done • Line 312: "... and both may..." (both what?) maybe
"... and both values of the GPH may..." rewritten for clarity • Line 321: "measure" !
"measurements" Done • Line 327: "Fig. 11" ! I think you mean Fig. 12. If not: Fig 12
is not referenced anywhere else in the text." Done • Line 348 to 350: Could you plot
this correlation? The correlation coeffs are: [0.909405 0.965083 0.984190 0.994004],
respective for the days. • Line 358: "... ozone units..." ! "... ozone number densities
to mixing ratios."Done • Line 373: "attitude" ! "altitude" Done • Line 374: "... is
also not..." ! "... is not..." Done • Line 376: "... as we have seen with the SNPP
spacecraft." - Do you have sources for this? What are you referring to here? GLEN
• Line 382: "... precision also within..." ! "... precision within..." Done • Line
386/387: "... of the atmospheric pressure vertical profile..." ! "... of the vertical profiles
of pressure and temperature..." Done • Line 388: "... as both well as..." ! "... as well
as..." Done • Line 391: "... that ARRM is capable of multi-year trend..." ! "... that
ARRM is capable of deriving multi-year trend..." Done

Figures • All axes should have a unit description or (AR) for arbitrary unit. Done.
• All unit descriptions should be uniform. So either choose "(unit)", e.g. in Fig. 4

"Altitude (km)", or ", unit" as in Fig. 1 "Altitude, km" Done. • All axes should have the same spelling for the label (e.g., Latitude vs. latit)Done. • I would suggest to write numbers below 13 as words ("one" instead of "1") Done for some numbers... • Figure 1: "normalized to 40.5 km" ! "normalized at 40.5 km" Done. • "field of view and includes no aerosols." !"field of view without aerosols." Done. • "is caused by" ! "originates from the" Done. • "tangent point starts" ! "tangent point start" Done. • "varies by 8-10%/km" With respect to what? Is it variation for the whole dataset? Done. • Figure 3: Maybe show the dependency on the scattering angle, perhaps by in- cluding an axes for the scattering angle? Done • Figure 4: "no aerosols as a function of altitude" Do you mean no aerosols? Or altitude independent aerosol extinction? Please clarify. Done. • Figure 6: Altitude != TH/Elevation - I think both terms are mixed up in the y axes description. It would also help to include a more detailed figure description on west/center/east slits as the meaning became clear only after reading the main text. Done. • Figure 8: Is this really the tropopause or just the 380 K isentropic surface? The tropopause line is from GMAO data. • Figure 11+12: Figure description is the same as in the main text. I suggest to reformulate the figure description. Done.

Please also note the supplement to this comment:
http://www.atmos-meas-tech-discuss.net/amt-2016-103/amt-2016-103-AC2-supplement.pdf

**Supplement:**

**Altitude Registration of Limb-Scattered Radiation**

Leslie Moy[1], P.K. Bhartia[2], Glen Jaross[2], Robert Loughman[3], Natalya Kramarova[1], Zhong Chen[1],

Ghassan Taha[4], Grace Chen[1], and Philippe Xu[5]

[1] Science Systems and Applications, Inc. (SSAI), 10210 Greenbelt Road, Suite 600, Lanham,

Maryland 20706  USA

[2] NASA Goddard Space Flight Center, Greenbelt, Maryland, USA

[3] Hampton University, Hampton, Virginia  USA

[4] GESTAR, Columbia, MD USA

[5] Science Applications International Corporation (SAIC), Lanham, MD

Correspondence to: Leslie Moy (leslie.moy@ssaihq.com)

**Abstract**

One of the largest constraints to the retrieval of accurate ozone profiles from UV backscatter limb sounding sensors is altitude registration. Two methods, the Rayleigh Scattering Attitude Sensing (RSAS) and Absolute Radiance Residual Method (ARRM), have been developed to determine the altitude registration to the accuracy necessary for long-term ozone monitoring. The methods compare model calculations of radiances to measured radiances, and are independent of onboard tracking devices. RSAS determines absolute altitude errors, but because the method is susceptible to aerosol interference, it is limited to latitudes and time periods with minimal aerosols. ARRM can be applied across all seasons and altitudes. However, it is only appropriate for relative altitude error estimates. The application of these methods to Ozone Mapping and Profiler Suite (OMPS) Limb Profiler (LP) measurements showed that, at launch, the OMPS LP instrument had a 1-2 km altitude registration error, resulting in a 50% error in the derived ozone density at some altitudes. Though some of the error has been attributed to thermal shifts in the focal plane of the instrument, most of it appears to be due to misalignment of the spacecraft star trackers or the OMPS LP focal plane with respect to the spacecraft axes. In addition, there are ±200 m seasonally varying errors that could either be due to errors in the spacecraft pointing information or in the geopotential height (GPH) data that we use in our analysis.

**Keywords: altitude registration, OMPS Limb Profiler, RSAS, ARRM, ozone profile, backscattered ultraviolet**

**1    Introduction**

Instruments that measure the solar radiation scattered by the earth's atmosphere in the limb direction provide a low cost way of measuring stratospheric ozone and aerosol profiles from satellites. The technique provides daily full coverage of the sunlit earth from commonly used polar sun-synchronous satellites. To meet long-term ozone monitoring needs (3% precision between 15 and 50 km) requires the altitude registration of the radiances to be accurate to within ~100 m. For a sensor orbiting at 800 km, this translates into ~6 arcsec accuracy in the pointing direction of the instrument line-of-sight (LOS) with respect to Earth's horizon.  This is often a difficult goal to achieve.

In this paper we critically examine the performance of two methods of altitude registration that compare measured and simulated radiances. We discuss the inherent strengths and limitations of each method and then assess their performance using data from the OMPS Limb Profiler (LP), launched onboard the Suomi NPP (SNPP) satellite on October 28, 2011.

One of these techniques, known as Rayleigh Scattering Attitude Sensing (RSAS), is relatively insensitive to instrument radiometric errors and drift. However, since the 350 nm/20 km limb radiances are greatly affected by aerosols, it works best where there is minimal aerosol loading. Under these condistions the accuracy of the method is limited 
[revised manuscript text omitted]

0.5% provided the OMPS TH is accurate to 1 km.  A second iteration of ARRM will remove any vestige of ozone sensitivity.

The main difficulty in applying ARRM is the inaccuracy of GPH data near 0.1 hPa needed to calculate 295 nm radiances at 65 km. To reduce this error we developed a variation of a technique that has been used for many years to derive mesospheric temperature profiles from the vertical slope of Rayleigh-scattered radiances measured by ground-based UV lidars (McGee et al., 1991). Temperatures were computed using the relative density differences between successive altitudes where the scattering mechanism was purely Rayleigh. Since our technique relies on this region of Rayleigh dominance, their technique can be applied to GPH (which is related to temperature assuming hydrostatic balance). The 350 nm and 295 nm residuals are affected similarly by the errors in the GPH with altitude so we use the 350 nm residual to correct for the GPH errors at 295 nm. Similiarly to the extent that stray light is wavelength independent, this correction will correct for stray light.

The residual at wavelength λ at altitude z, defined as $d(\lambda,z)= \ln I_m(\lambda,z)-\ln I_c(\lambda,z)$, is corrected using 350 nm residuals:

$$d_{corr}(\lambda, z) = d(\lambda, z) - [d(350, z) - d(350, z_0)] \qquad (2)$$

where $z_0$ is a normalization altitude.

The 350 nm differential residuals on the right side of equation (2) provide an estimate of the relative error in calculating radiances using meteorological data between z and $z_0$. Since this error should be wavelength independent, we can use this term to correct the residuals at any wavelength, assuming that the meteorological data at $z_0$ is accurate and that the 350 nm wavelength is well calibrated. The large response of OMPS LP at 350 nm results in signals that are the least affected by out-of-band stray light.

The TH error estimated using this method is given by:

$$\Delta z = \frac{d_{corr}(\lambda,z)}{s(\lambda,z)-[s(350,z)-s(350,z_0)]} \qquad (3)$$

We are minimizing ozone profile sensitivity by applying this method to radiances at wavelengths shorter than 300 nm. At longer wavelengths DUR makes the LS radiances sensitive to total column ozone at all altitudes. At 295 nm, the use of z near 65 km provides low ozone sensitivity.

Though it is best to set $z_0$ as low as possible to minimize GPH caused errors, aerosol
contamination limits the value to around 40 km.

ARRM has two primary uncertainties. Since 1% error in radiance calibration produces ~70 m
error in determining the TH, this method requires accurate radiances (or sun-normalized
radiances) and may be affected by instrument degradation. Though the absolute accuracy of
ARRM may not be as good as RSAS, this method can be applied at latitudes/seasons where
RSAS cannot be applied reliably because of aerosol contamination. And like RSAS, this method
is also sensitive to errors in GPH profile near 3 hPa, which are used for calculating 350 nm
radiances at 40 km.

**2.3    "Knee Method"**

The name of this method is derived from the characteristic knee shape of the limb radiance
profiles (Fig. 4). Above the knee the radiances decrease with altitude due to exponential decrease
in Rayleigh scattering and ozone density. Below the knee ozone absorption becomes so large that
it essentially blocks most of the Rayleigh-scattered radiation from reaching the satellite, making
the radiances insensitive to atmospheric pressure. This characteristic shape allows estimations of
altitude registration error in a manner very similar to that of RSAS. An advantage of this method
is the ability to use shorter wavelengths which are less sensitive to aerosols. However, 
[revised manuscript text omitted]
. There is better agreement at the poles, but this may be due to the reliance on climatology where there are scant measurements. As a result, it is not clear how these GPH errors influence the ARRM results. However, in Section 4.3 we discuss some suggestive but inconclusive results untangling GPH errors from TH errors.

**Section 4.2    Radiances comparison**

We previously described (in Section 2) the difficulty modeling DUR caused by scene heterogeneity and aerosols. Both the RSAS method and ARRM depend upon an accurate model for DUR at 40 km relative to other altitudes, and any model errors translate directly (Equations 1 and 3) into false estimates in the TH errors.

We estimate the DUR modeling error by comparing LP measurements and modeled 353 nm radiances at 3 hPa. The OMPS Nadir instrument makes nearly simultaneous measurements from a smaller field-of-view (50 x 50 km at nadir).The surface reflectivity derived from its 340 nm radiances are therefore relatively insensitive to DUR effects (compared to measurements derived from the LP measurements). With better reflectivity assumptions the model/measurement comparisons offer a lower bound of the effect of DUR modeling errors.

The radiance comparison, shown in Fig. 12, suggests model or calibration errors of 2-3% on average, plus structures caused by the limb and nadir scene mismatch. If this error were attributed solely to the limb modeled DUR effect, the resulting TH error would be less than +/-

200 m. There is no evidence of either a seasonal or a latitude dependence in the four days of comparisons, meaning that DUR effects cannot explain the robust seasonal and latitudinal variations seen in ARRM results (Fig. 9 and Fig. 10). These model/measurement comparisons provide an estimate of errors related to incomplete modeling of DUR and inhomogeneous surface albedo included in our RSAS and ARRM results. We therefore conclude those variations arise from errors in the GPH scale or from true TH variations.

**Section 4.3    Ozone comparison**

At 3 hPa limb ozone retrievals are very sensitive to TH errors, with 20 to 25% per km change in ozone concentration (see Fig. 5). Similar to the Knee Method, we can use this sensitivity to gauge the residual TH errors. We compare LP ozone retrievals against Aura MLS v4 ozone retrievals at 3 hPa (near 40 km) (Fig. 13). While the latitudinal patterns of differences significantly vary with season, we find agreement within ±10% over all seasons and latitude bands. If completely interpreted as due to TH error, a 10% difference would translate to less than

500 m error. These comparisons confirm a residual uncertainty in our scene-based altitude registration techniques of ±200m.

The ARRM method has displayed the ability to track any drifts or sudden changes of 50 m (Sect.

3.3), and time series of TH error derived from the ARRM method  track very closely to the time series of the LP/MLS 3 hPa ozone differences (Fig. 14). The highest correlation (0.76) was found at 45° south latitude, with considerable smaller values in the northern hemisphere (0.30 at 60°

north).  Whether this suggests the ARRM results can be attributed solely to TH errors has not been determined yet.

Both ARRM and LP/MLS ozone comparison depend upon accurate TH and MERRA

information, and in the same way.  So, while these results suggest some confidence in the

ARRM technique, we cannot assign the correlation shown in Fig. 14 to only a TH error or a

MERRA error. It is important to note that MLS ozone profiles are reported as volume mixing
ratio on a vertical pressure grid, while the LP algorithm retrieves ozone as number density on an
altitude grid. Thus, in order to compare LP and MLS ozone retrievals we had to convert ozone
number densities to mixing ratios using MERRA temperature and GPH profiles. This conversion
inevitably introduces errors in MERRA GPH and temperature into the ozone comparisons.
Therefore ozone differences between LP and MLS ozone retrievals not only depend on the LP
TH error, but on errors in MERRA GPH as well as on errors in the retrieval algorithms and
instrumental sampling (geophysical noise). Furthermore, analysis of LP and MLS ozone
retrievals indicates a large daily ozone variability within a 5-degree latitude bin at 3 hPa that
ranges from 2% in the tropics to 20% at high latitudes with the seasonal maximum during austral
winters (results are not shown here), which can give readers a sense of geophysical ozone
variability. In consideration of all of the above factors, we remain cautious in making definite
conclusions and applying time-dependent corrections for the LP TH at this time; further analysis
and comparisons with independent ozone observations (like SAGE III) are needed to confirm the
results.

**Section 5      Conclusions**

Accurate altitude registration is key to the success of the limb scattering measurement
technique. We have described two scene-based techniques that together provide highly precise
and accurate estimates of the tangent height. These altitude registration techniques are
inexpensive and more comprehensive than external sources of altitude information, such as star
trackers mounted on the spacecraft. Star trackers are typically used on spacecraft when accurate
pointing knowledge is desired, but this accuracy does not necessarily transfer to the limb
measurements. In Section 3.1 we've described pointing shifts that occur within the sensor optics.
Though we were able to calibrate thermal sensitivities within the OMPS instrument, we have yet
to identify the source of 1-1.5 km pointing errors derived from RSAS (see Table 1). These may
arise from mounting  offsets  of the instrument and star trackers, or from spacecraft flexure
between the two.
The RSAS and ARRM techniques are complementary because the former is
accurate to ±200 m, but only under limited conditions. The accuracy of
ARRM cannot be easily established, but it has a precision within

±200    m. We    believe    this    results    in    small,    less    than    100    m,    trend
uncertainties    for    sufficiently    long    time    series,    as    demonstrated    by    the
OMPS ARRM record.

The single largest source of uncertainty in both techniques is knowledge of the vertical profiles
of pressure, which must be provided from external sources. Given uncertainties in GPH data, as
seen in the MLS comparison, as well as uncertainties in our ozone retrieval algorithm (not
related to TH error), it is currently not possible to tell if the latitudinal and seasonal variations
seen in ARRM results are caused by TH error. Further work will be needed to understand their
cause. We have shown, however, that ARRM is capable of deriving multi-year trend
uncertainties that are on the order of 100m or smaller. Furthermore, the two TH registration
methods discussed in the paper allow us to track any drifts or sudden changes in our altitude
registration to better than 50 m, which is the minimum level necessary to derive accurate ozone
trends from a limb technique.

Acknowledgements: The authors gratefully acknowledge the assistance of NASA's Limb
Processing Team in providing the data used in this paper.  We would also like to thank Dave
Flittner, Ernest Nyaku and Didier Rault, helped with the development and updates of the RT
model. Finally, we'd like to acknowledge the role Didier played in laying the groundwork for the
OMPS limb retrieval algorithm

Table 1: RSAS results at the equator before the Kelud eruption 2014 February. The time period had a minimum value (during OMPS life time) and was chosen using OSIRIS measurements (Fig.8).

| TH error, km | EAST | CENTER | WEST |
|---|---|---|---|
| RSAS results | 1.12 | 1.37 | 1.52 |

[Figure]

**Figure 1:** Figure 1a shows calculated 350 nm radiances as a function of altitude, normalized at
40.5km. The calculation models the OMPS LP field of view without aerosols.  The shape of the
curve originates from the competition between molecular scattering, which increases roughly
linearly with pressure, and attenuation which becomes important when the Rayleigh optical
thicknesses near the tangent point start to become large. Attenuation causes the slope of 350
nm radiances to change sharply between 40 and 20 km (Fig. 1b), a ~9% /km difference between
and 40 km. The slope is used to estimate altitude registration errors by comparing measured
ratios with model simulated ratios.

[Figure]

**Figure 2:** The 353 nm sun-normalized radiances from one orbit of OMPS LP (central slit) taken on Feb. 2, 2012. The blue line shows 40.5 km values and the green line shows 20.5 km values (divided by 8 to put both curves on a similar scale) versus latitude. Since the global aerosol loading on this day was small, the short scale features in both curves are largely caused by variations in cloud and surface albedo. The 20.5 km curve has sharper features and appears to be shifted towards toward the South Pole. This is because large Rayleigh attenuation at 20.5 km causes the radiances to have much higher sensitivity to the atmosphere on the satellite side of the tangent point (TP), while 40.5 km radiances have similar sensitivities to both sides. This effect creates large noise in applying the RSAS technique to orbital data. However, since the noise varies randomly from orbit to orbit, it can be reduced by averaging data from multiple orbits. Figure 6 of Loughman et al. (2015) is an example of how the contributions become asymmetric about the tangent point at lower THs.

[Figure]

**Figure 3:** The ratio of 353 nm limb-scattered radiances at 20.5 km with and without aerosols
(left axis) and single scattering angle (right axis) as a function of latitude. A nominal latitude-
independent aerosol extinction profile was used in the calculation for the OMPS LP viewing
geometry on Feb 2, 2012. The strong latitude dependence is caused by an order of magnitude
change in aerosol scattering phase function with latitude combined with the attenuation of
Rayleigh-scattered radiation by aerosols along the line-of-sight (LOS). In the southern
hemisphere, where LP measures aerosols in the backscatter direction, the latter effect
dominates and the radiation decreases. The net effect is very sensitive to altitude, variation of
aerosol extinction profile along the LOS, and aerosol particle size distribution, and is therefore
difficult to calculate accurately.

[Figure]

**Figure 4:** Figure 4a shows calculated 305 nm radiances as a function of altitude assuming an atmosphere with no aerosols. The slope (Fig. 4b) is caused by competition between Rayleigh scattering and ozone absorption near the altitude of maximum radiance, ~44 km. Above 55 km the sensitivity is nearly constant in height, ~13%/km at 65 km. Above the knee the radiances decrease with altitude due to the exponential decrease in Rayleigh scattering and ozone density. Below the knee the ozone absorption becomes so large that it essentially blocks most of the Rayleigh-scattered radiation from reaching the satellite, making the radiances insensitive to atmospheric pressure. This characteristic knee shape allows one to estimate altitude registration error in a manner very similar to that of RSAS, but also makes it is very susceptible to ozone profile assumptions, as illustrated in Fig. 5.

[Figure]

**Figure 5:** Typical ozone profile in the tropics (such as the one shown in the left panel) peak
between 25 and 30 km. By shifting the ozone profile we can estimate an order and pattern of
error in ozone profiles due to TH shift (right panel). Errors in ozone retrievals are within 8% at
40 km from TH errors of 300 m. Errors are least sensitive at the ozone peak, and are more
variable below.

[Figure]

**Figure 6:** OMPS LP CCD high gain earth viewing radiance images for the three slits (east/center/west). The east and west slit images are separated in longitude by 2.25° from that of the center slit. The wavelength range for each image is 270 to 1050 nm and the minimal TH range is 0 to 80 km. The CCD has 740 pixels in the wavelength dimension. There are 340 pixels in the spatial dimension; the high gain images occupy the lower half of the CCD (pixels 0 to 170). The spatial extent of each slit's image on the detector is limited by the vertical length of that slit. The lower slit edges (nearest the Earth surface) provide a high contrast signal cutoff that can be monitored for movement.

[Figure]

**Figure 7:** Slit edge results for the three slits (Green=East Slit, Red=Center Slit, Blue=West Slit)
plotted against time since Southern Terminator crossing. A 1 pixel shift corresponds to a 965 m
TH shift. The offsets are stable from the southern terminator to the mid latitude northern
hemisphere where the exposure to the sun increases thermal effects.

[Figure]

**Figure 8:** Time series of OSIRIS aerosol extinction profiles above the tropopause (dashed line).
The large concentration in 2012 is due to the June 2011 Nabro eruption in Eritrea. The aerosols
at 20 km reached a minimum value (during OMPS life time) just before the eruption of the
Kelud Volcano on 14 February 2014.

[Figure]

**Figure 9:** Time dependent plots of TH errors from ARRM analysis at five latitude bands for the
three slits. Values are normalized at the Equator just prior to the Kelud eruption on February
14, 2014 based on the RSAS results summarized in Table 1. Arrows indicate a 12 arcsec pointing
adjustment to one of the two spacecraft star trackers on April 25, 2013. The resulting 100 m TH
shift can be seen most clearly by comparing 2012 and 2013 results. Slit discrepancies and
seasonal dependencies of +/-200 m can also be seen.

[Figure]

**Figure 10:** Average (over the ~4 year study period) ARRM results by latitude and seasons
(MAM-green, JJA-red, SON-purple, DJF-blue) for Center Slit. Excluding the extreme polar
regions, there appears to be an average 300 m TH change over an orbit.

[Figure]

**Figure 11**: Daily five degree zonal means of GPH from MLS (blue), GMAO (green), and the difference MLS-GMAO (red) at 3 hPa GPH for four cardinal days. Note that despite a 2 to 4 km change over an orbit, the differences are generally within 200 m. These differences provide an estimate of the errors caused by the use of MERRA GPH in our radiative transfer calculations. Better agreement seen at the poles may simply be due to the fact that there are not many measurements at these latitudes and both may be influenced by the same climatology. If this error were attributed solely to the limb model and only at one altitude, the resulting TH error would be less than +/-200 m. There is no evidence of either seasonal or latitude dependence in the comparison, meaning that DUR effects cannot explain the variations seen in Fig. 9.

[Figure]

**Figure 12:** We have estimated the DUR modeling error by comparing 353 nm measured and
modeled radiances at 3 hPa. The radiances are modeled using an independent, nearly
simultaneous measure of surface reflectivity derived from the OMPS Nadir instrument at 340
nm. The 50x50 km nadir-view measurements are relatively insensitive to DUR effects. The
radiance differences (given for the same four cardinal days as in Fig. 11) suggest model or
calibration errors of 2-3% on average, plus structure caused by the contributing limb and nadir
scene mismatch.

[Figure]

**Figure 13:** Daily five degree zonal means of ozone from MLS (blue), GMAO (green), and the
MLS-GMAO differences (red) at 3 hPa GPH for four cardinal days. The differences are generally
within 6% which if completely attributed to TH error would be ~200 m.

[Figure]

**Figure 14:** The time series of daily zonal mean ozone differences (%) between OMPS LP and
Aura MLS for the three slits (top). OMPS LP profiles are corrected with the TH error shown in
Table 1. Note the similarity in time-dependent patterns for ozone differences and  TH error
derived from AARM method (bottom, reprinted from Fig.9). The fact that these two completely
independent methods show very similar patterns give us additional confidence in the AARM
method.

---

## Referee Comment (RC2) · Anonymous Referee #1 · 15 Jul 2016

This paper provides an important analysis of the altitude registration of the OMPS Limb Profiler data set. This is a critical and difficult aspect of limb sounding data sets and is motivated by the steep accuracy and precision requirements on ozone monitoring that make it especially relevant to OMPS. Several different methods are discussed and a few are applied to the OMPS data set. The results are important and it seems that the authors have made good progress with understanding the corrections that need to be applied to the OMPS data; however, the paper is very short on detail and makes substantial claims that are not justified by the analysis. The overall logical flow of the paper is hard to follow even though the methods and analysis are quite simple. For example, it is not clear which corrections are applied and in what order. Is the slit correction applied or just analysed? Then it seems that the RSAS analysis is applied as a single value across the entire data set and then ARRM is investigated but not applied as the uncertainties are not understood. Yet, the conclusions state that the ARRM-corrected OMPS record has an altitude registration error of less than 50 m,

which is substantially better than any previous studies would suggest is possible with methods such as these. In my opinion, the paper has potential to be published in AMT, but revision is required, most importantly a re-working of the logical flow of the paper and the accompanying explanations.

New information including results and discussion is provided in the figure captions that is not provided or discussed in the text. This may be part of the problem that makes the logic hard to follow. This is should be revised and all statements comprehensively discussed in the main body of the paper (Fig 2, Fig 4, Fig 11, Fig 12 are extreme examples of this.) Additionally, the section on the uncertainties is especially hard to follow and needs reworking.

Abstract: The Knee method should also be mentioned as it has been used extensively in the past and the current abstract makes it sound like there are only two methods (RSAS and ARRM)

Abstract, line 36: This statement seems to insinuate that the authors developed the RSAS method and that it is specific to ozone monitoring.

Introduction, line 65: many other species beyond ozone and aerosol can also be measured by limb scattering, even though OMPS produces only these two products due to spectral resolution limitations, other instruments have produced many other products.

Introduction, line 68: the 100 m accuracy requirement needs a reference

Introduction, line 78: statement regarding insensitivity of RSAS to radiometric errors and drift needs justification or at least a reference

Theoretical Basis, line 90: The claim is that "most scene-based altitude registration methods" use the gradient in the Rayleigh scattering profile. Are there some that don't? Are there only the three mentioned later (RSAS, ARRM, Knee)?

Theoretical Basis, line 95: Clouds outside the "circular cone" from the tangent point to the horizon can impact the contribution of the upwelling radiance to the limb signal.

Theoretical Basis, line 142: At several points in the paper, the same statement is made that "RSAS method work best where aerosol extinction is small". This needs to be quantified. How small? How does this error relate to the errors that arise from the uncertainty in the pressure profiles? Or from the signal to noise levels in the measured radiances? Also, what about uncertainty in the upwelling radiance? Is it truly effectively zero in the 40/20 ratio? Sections 3.2 and 4.2 make conflicting statements about the importance of accurate modelling of the DUR.

Theoretical Bases, line 157: What is the reason that the sensitivity of the 295 nm radiance to ozone drops substantially above 65 km? Is this ozone in the line of sight, or total column?

Theoretical Basis, line 159: Did the authors develop ARRM, or is there a reference that should be cited? They begin this discussion about the problems with the method before stating how it works, or even any reference to the technique.

Theoretical Basis, line 165: It is unclear what this statement about the goal of "correcting" radiance residuals means in the context of deriving the tangent altitude information.

Theoretical Basis, line 186: Again, quantification of these statements is necessary. "ARRM may not be as good as RSAS" – in what conditions and how by how much? At what value of aerosol extinction, or conversely at what uncertainty in absolute radiance, would you prefer to use ARRM over RSAS or vice versa?

Theoretical Basis, line 201: It is not clear that shifting the ozone profile as is done in Fig. 5 translates to exactly the same error in altitude registration in the UV limb radiance. Also, the discussion/analysis surrounding Fig. 5 neglects the non-linearity of the inversion, which is especially important below 20 km, i.e. shifting the registered tangent altitude and performing the retrieval does not produce identical results to simply shifting the ozone profile by that same amount.

Validation: The logical tracing of the errors coming from the different terms is very

difficult to follow and this entire section is in need of some systematic thought and re-organization.

Validation, line 358: It does not appear that the cross referenced Section 3.3 results support the claim that ARRM can "track drifts or sudden changes of 50 m"

Conclusion, line 400: What is the OMPS ARRM record? The statement was made that the ARRM results are not applied to the OMPS data.

Conclusion, line 401: Is the uncertainty in assumed pressure profiles the biggest source of uncertainty in both methods? Previously it was explained that it is aerosol for RSAS...?

Conclusion, line 409: It is not convincingly shown that these methods provide altitude registration tracking to better than 50 m.

Figure 1: The statement in the caption: "Since the ratio of 40 to 20 km radiances at 350 nm varies by 8-10%/km..." is confusing. How can the ratio of radiances at two set tangent altitudes vary with tangent altitude?

Figure 2: Why 353 nm instead of 350 as in Fig. 1?

Figure 3: This should be shown together with the solar scattering angle at the tangent point so it is clear what is happening here (for readers not familiar with OMPS this will not be obvious). Also the authors should consider showing some of the complexity involved with the aerosol parameters by repeating this same curve for different altitudes, extinction profiles, and particle size distributions, all of which are stated as important factors.

Figure 9: Are these results for individual radiance profiles, or daily averages? The low amount of noise is surprising based on previous studies using such methods as these.

[Figure]

---

## Author Comment (AC3) · 6 Sep 2016

Replies are indicated with '»>' : Anonymous Referee #1 This paper provides an important analysis of the altitude registration of the OMPS Limb Profiler data set. This is a critical and difficult aspect of limb sounding data sets and is motivated by the steep accuracy and precision requirements on ozone monitoring that make it especially relevant to OMPS. Several different methods are discussed and a few are applied to the OMPS data set. The results are important and it seems that the authors have made good progress with understanding the corrections that need to be applied to the OMPS data; however, the paper is very short on detail and makes substantial claims that are not justified by the analysis. The overall logical flow of the paper is hard to follow even though the methods and analysis are quite simple.

For example, it is not clear which corrections are applied and in what order. Is the slit correction applied or just analysed? Then it seems that the RSAS analysis is applied

as a single value across the entire data set and then ARRM is investigated but not applied as the uncertainties are not understood. Yet, the conclusions state that the ARRM-corrected OMPS record has an altitude registration error of less than 50 m, which is substantially better than any previous studies would suggest is possible with methods such as these. In my opinion, the paper has potential to be published in AMT, but revision is required, most importantly a re-working of the logical flow of the paper and the accompanying explanations. New information including results and discussion is provided in the figure captions that is not provided or discussed in the text. This may be part of the problem that makes the logic hard to follow. This is should be revised and all statements comprehensively discussed in the main body of the paper (Fig 2 Fig 4 Fig 11 Fig 12-updated are extreme examples of this.) Additionally, the section on the uncertainties is especially hard to follow and needs reworking.

Abstract: The Knee method should also be mentioned as it has been used extensively in the past and the current abstract makes it sound like there are only two methods (RSAS and ARRM). »>Our paper focuses on describing on the two techniques that we use to determine altitude registration for OMPS (RSAS and ARRM), and results from these techniques. Although we describe the Knee method in the body of the paper for historical reasons, we do not use it and we therefore believe it does not warrant a mention in the Abstract.

Abstract, line 36: This statement seems to insinuate that the authors developed the RSAS method and that it is specific to ozone monitoring. »> New text: Two methods, the Rayleigh Scattering Attitude Sensing (RSAS) and Absolute Radiance Residual Method (ARRM), are able to determine altitude registration to... Also line 122: Though several variations of the RSAS technique have been developed for ozone sensors (McPeters et al., 2000; Rault et al., 2005; Taha et al., 2008), we find ... Introduction, line 65: many other species beyond ozone and aerosol can also be mea- sured by limb scattering, even though OMPS produces only these two products due to spectral resolution limitations, other instruments have produced many other products. »>New

text: Instruments that measure the solar radiation scattered by the earth's atmosphere in the limb direction provide a low cost way of measuring trace gases, aerosol profiles and clouds from satellites.

Introduction, line 68: the 100 m accuracy requirement needs a reference »>Reference included

Introduction, line 78: statement regarding insensitivity of RSAS to radiometric errors and drift needs justification or at least a reference. »> New text: One of these techniques, known as Rayleigh Scattering Attitude Sensing (RSAS), is relatively insensitive to instrument radiometric errors because it utilizes measurements at two altitudes (20 and 40 km) where many of the errors are correlated.

Theoretical Basis, line 90: The claim is that "most scene-based altitude registration methods" use the gradient in the Rayleigh scattering profile. Are there some that don't? Are there only the three mentioned later (RSAS, ARRM, Knee)? »>We used the word 'most' because even though we did a literature search and only found references to variations of the RSAS and Knee methods, we leave open the possibility there are others.

Theoretical Basis, line 95: Clouds outside the "circular cone" from the tangent point to the horizon can impact the contribution of the upwelling radiance to the limb signal. »>New Text: For wavelengths longer than 310 nm, the limb-scattered radiance has a significant contribution from diffuse upwelling radiance (DUR), which is affected by tropospheric clouds, aerosols and surface albedos within a circular cone whose base extends hundreds of km to the horizon. The further these reflectors are from the apex of the cone, the less is their contribution to DUR. At non-ozone absorbing wavelengths DUR can be as much as half of the measured radiance. Since DUR is challenging to model accurately, all successful altitude registration methods must be relatively insensitive to reflectivity variations within the cone.

Theoretical Basis, line 142: At several points in the paper, the same statement is made

that "RSAS method work best where aerosol extinction is small". This needs to be quantified. How small? How does this error relate to the errors that arise from the uncertainty in the pressure profiles? Or from the signal to noise levels in the measured radiances? »>We use RSAS when/where the aerosol effect is minimal (ideally none but that's hard to determine). On average (using an aerosol climatology the effect is ~1% (100 m) in the southern hemisphere and sharply increasing to ~2% (200 m) in the north pole), but averages can be deceiving. Our strategy was to find the cleanest time/place according to OSIRIS measurements (Fig. 8). There is new text in Section 3.2. Aerosol errors are independent of pressure errors so the two will sum in quadrature.

Also, what about uncertainty in the upwelling radiance? Is it truly effectively zero in the 40/20 ratio? Sections 3.2 and 4.2 make conflicting statements about the importance of accurate modelling of the DUR. »>New Text Section 2.1 (line 170): However, since the noise varies randomly from orbit to orbit, DUR modeling errors are reduced by averaging data from multiple orbits (this is confirmed in daily averages of the sun-normalized radiances where short scale features are not seen). Also our estimate of our RSAS uncertainty is based on both the southern polar region and the equator which span the range of surface heterogeneity (the source of DUR variation). New text in Section 3.2

Theoretical Bases, line 157: What is the reason that the sensitivity of the 295 nm radiance to ozone drops substantially above 65 km? Is this ozone in the line of sight, or total column? »>Text added (Section 2.2): Though 295 nm radiances can be very ozone sensitive, this sensitivity drops to less than 0.2% for a 10% change in ozone above 65 km because the ozone density at high altitudes is exceedingly low. (see Figure 5).

Theoretical Basis, line 159: Did the authors develop ARRM, or is there a reference that should be cited? They begin this discussion about the problems with the method before stating how it works, or even any reference to the technique. »>Sentence added to Abstract: "ARRM, a new technique introduced in this paper, can be applied across

all seasons and altitudes." And Section 2.2: We developed ARRM to be applicable over many latitudes and times.

Theoretical Basis, line 165: It is unclear what this statement about the goal of "correcting" radiance residuals means in the context of deriving the tangent altitude information. »>Paragraph reworded to distinguish between McGee's findings and our method.

Theoretical Basis, line 186: Again, quantification of these statements is necessary. "ARRM may not be as good as RSAS" – in what conditions and how by how much? At what value of aerosol extinction, or conversely at what uncertainty in absolute radiance, would you prefer to use ARRM over RSAS or vice versa? »> It is a question of absolute TH error versus relative error. Because RSAS uses ratios it cancels out much of the instrument errors. The sentence has been rewritten to reflect this.

Theoretical Basis, line 201: It is not clear that shifting the ozone profile as is done in Fig. 5 translates to exactly the same error in altitude registration in the UV limb radiance. Also, the discussion/analysis surrounding Fig. 5 neglects the non-linearity of the inversion, which is especially important below 20 km, i.e. shifting the registered tangent altitude and performing the retrieval does not produce identical results to simply shifting the ozone profile by that same amount. »>True. We now say it is an estimate and explicitly name the qualifiers you point out.

Validation: The logical tracing of the errors coming from the different terms is very difficult to follow and this entire section is in need of some systematic thought and re-organization. »>Rewritten section with our reasoning better described.

Validation, line 358: It does not appear that the cross referenced Section 3.3 results support the claim that ARRM can "track drifts or sudden changes of 50 m" »>We addedd a figure (new Figure 10) and changed the value.

Conclusion, line 400: What is the OMPS ARRM record? The statement was made that the ARRM results are not applied to the OMPS data. »> We were referring to the

ARRM timeseries in Figure 9. The ARRM results showed errors within our uncertainty so were not applied.

Conclusion, line 401: Is the uncertainty in assumed pressure profiles the biggest source of uncertainty in both methods? Previously it was explained that it is aerosol for RSAS. In the southern hemisphere, the GPH is the larger error source. It is difficult to determine which error source is larger in the northern hemisphere, but we believe we have been conservative in our error estimations to include the full range of variations.

Conclusion, line 409: It is not convincingly shown that these methods provide altitude registration tracking to better than 50 m. »>The conclusion section has been rewritten.

Figure 1: The statement in the caption: "Since the ratio of 40 to 20 km radiances at 350 nm varies by 8-10%/km" is confusing. How can the ratio of radiances at two set tangent altitudes vary with tangent altitude? »> Attenuation causes the slope of 350 nm radiances to change sharply between 40 and 20 km (Fig. 1b), a ~10% /km difference between 20 and 40 km.

Figure 2: Why 353 nm instead of 350 as in Fig. 1? fixed

Figure 3: This should be shown together with the solar scattering angle at the tangent point so it is clear what is happening here (for readers not familiar with OMPS this will not be obvious). Also the authors should consider showing some of the complexity involved with the aerosol parameters by repeating this same curve for different altitudes, extinction profiles, and particle size distributions, all of which are stated as important factors. »> Scattering angle included in figure.

Figure 9: Are these results for individual radiance profiles, or daily averages? The low amount of noise is surprising based on previous studies using such methods as these. »>Yes, these are daily averages – text updated

[Figure]